# Working Capital Management Policies in Indian Listed Firms: A State-Wise Analysis

**Najib H. S Farhan [1], Faozi A. Almaqtari [2] , Ebrahim Mohammed Al-Matari [3,4,* ] , Nabil Ahmed M. SENAN [5,6], Waleed M. Alahdal [7,* ] and Saddam A. Hazaea [8]**

1 Department of Accounting, Faculty of Management Sciences, Ibb University, Ibb 40440, Yemen; najib720000@gmail.com

2 Department of Accounting, College of Commerce and Economics, Hodeidah University, Hodeidah 1814, Yemen; fouzi_gazim2005@yahoo.com

3 Department of Accounting, College of Business, Jouf University, Al-Jouf 75911, Saudi Arabia

4 Faculty of Commerce and Eonomics, Amran University, Amran 83736, Yemen

5 Department of Accounting, College of Business Administration, Prince Sattam Bin Abdul Aziz University, Al Kharj 11942, Saudi Arabia; nabil_senan@yahoo.com

6 Accounting Department, Administrative Science College, Al-Baydha University, Al-Baydha R621, Yemen

7 Department of Accounting, Faculty of Business, Economics and Social Development, Universiti Malaysia Terengganu, Kuala Nerus 21030, Malaysia

8 School of Accounting, Yunnan University of Finance and Economics, Kunming 650221, China; sadhi792@gmail.com

* Correspondence: emalmatri@ju.edu.sa (E.M.A.-M.); wm.alahdal2011@gmail.com (W.M.A.)

**Abstract:** The main aim of this paper is to evaluate the impact of working capital policies on firms' profitability. The study uses a panel data set of 829 manufacturing firms for the period from 2011 to 2017. Data is extracted from Prowess IQ database. An empirical model is used for testing research hypotheses. The results show that all firms across Indian states follow conservative financing and investment policy. The conservative investment policy positively affects return on assets, whereas the conservative financing policy negatively affects return on assets and therefore firms' financial sustainability. Regulators, policymakers, investors, and financial managers in Indian manufacturing companies are advised to follow a conservative investment and financing policy, which is effective and efficient in boosting firms' profitability for attaining financial sustainability. Therefore, manufacturing firms should invest more in current assets, because they need to expand both inventories and trade credit to their customers. Moreover, financial managers are advised to favor a low level of debt in financing assets. Apart from previous literature, which was either descriptive or based on a small sample size, the present study makes a novel and significant contribution by bridging an existing gap through applying a panel fixed- and random-effect model for a large sample: 829 firms. Furthermore, the business environment in India is somewhat different from that of other countries around the globe, which makes investigating working capital policies in the Indian contexts an interesting endeavor.

**Keywords:** Indian firms; financing policy; investing policy; state-wise analysis

## 1. Introduction

The Modigliani and Miller capital structure theory is based on the view that firm's value does not get affected by the method of financing inefficient markets where bankruptcy cost, taxes, cost of information, and agency cost is absent [1]. After some time, Smith [2] introduced a new thought by highlighting the effect of working capital on firms' performance sustainability such as firms' risk, value, and profitability. Working capital management has been gaining considerable importance because of its contribution toward enhancing financial sustainability and shareholders' value by making a proper tradeoff between risk and profitability [3–6]. Working capital is directly connected to the firm's liquidity and

profitability. It shows how healthy the financial position of a company is [7]. It is well known that the greater the risk, the higher profitability a firm achieves.

Firms can adopt either an aggressive working capital policy or a conservative working capital policy. When a company follows an aggressive policy, it keeps a low level of working capital and uses short-term sources for financing its requirements [8]. The lowest risk is always associated with the lowest return. This strategy is referred to as a conservative policy [9]. When a firm follows an aggressive asset management policy, it minimizes its capital held in current assets. Such firms may achieve high profitability, but at the same time, they are subject to high risk [10]. Empirical evidence indicated the importance of mixed financing policy of working capital for financial sustainability [11]. Quansah [12] argued that if firms adopt an aggressive working capital policy, they should balance it with a conservative working capital policy.

Chen and Kieschnick [11], Afza and Nazir [13], and Howorth and Westhead [14] argued that conservative working capital investment policy results in a huge amount of investment in current assets. However, more investment in current assets results in increasing insurance cost, inventory storage cost, and financial costs, which may reduce firms' profitability [13]. However, an aggressive working capital investment policy is associated with high risk and high profitability [15]. When a firm adopts an aggressive working capital financing policy, it increases the level of current liabilities [16].

Al-Shubiri [17] argued that there is a negative relationship between aggressive investment policy and Tobin's Q, while aggressive financing policy positively impacts Tobin's Q of firms listed at the Amman Stock exchange. On the other hand, Vahid et al. [18] revealed that conservative investment policy negatively impacts firms' profitability, while aggressive investment policy positively impacts firms' profitability. In contrast, Pestonji and Wichitsathian [19] found that working capital investment policy has a positive and significant impact on firms' profitability, whereas working capital financing policy negatively impacts firms' profitability.

The business environment is one of the factors that affects the adoption of a particular working capital policy. Business environment refers to all institutions, individuals and other forces that are not under the control of a business. The business environment in India is somewhat different from that of other countries around the globe, which makes investigating working capital policy in the Indian context an interesting research, giving special attention to all firms across Indian states. For instance, monetary policy may have a different impact across states. The reason is that, in such large federal structures, another dimension is introduced: federal state governments [20]. As long as the concept of the economic region is logically different from that of federal states, this encourages the authors to examine working capital policies adopted by firms across all Indian states.

Table 1 reveals that there are 29 states and 7 union territories. India is a secular, sovereign, and democratic republic based on the parliamentary system. The president is the head of the union. In each state, there is a governor who represents the president; the government system in each state is similar to that of the union.

The common model of a monetary policy anticipates a uniform impact of such policy on the country economy. Notably, this belief does not consider the fact that in some countries such as India and the USA, the economy is composed of different states that may respond differently to macroeconomic stimulations and changes. For instance, the impact of a change in food prices would be different in states that are the dominant producers of that kind of food as compared to states that are not. Likewise, monetary policy may have a different impact across states. The reason is that, in such large federal structures, another dimension is introduced; federal states' governments [20].

**Table 1.** The Indian states and union territories.

| No | State | No | State | No | State | No | State | No | State |
|----|-------|----|-------|----|-------|----|-------|----|-------|
| 1 | Andaman & Nicobar Islands | 8 | Dadra and Nagar Haveli | 15 | Jharkhand | 22 | Meghalaya | 29 | Rajasthan |
| 2 | Andhra Pradesh | 9 | Daman and Diu | 16 | Karnataka | 23 | Mizoram | 30 | Sikkim |
| 3 | Arunachal Pradesh | 10 | Goa | 17 | Kerala | 24 | Nagaland | 31 | Tamil Nadu |
| 4 | Assam | 11 | Gujarat | 18 | Lakshadweep | 25 | Nct of Delhi | 32 | Telangana |
| 5 | Bihar | 12 | Haryana | 19 | Madhya Pradesh | 26 | Odisha | 33 | Tripura |
| 6 | Chandigarh | 13 | Himachal Pradesh | 20 | Maharashtra | 27 | Puducherry | 34 | Uttarakhand |
| 7 | Chhattisgarh | 14 | Jammu and Kashmir | 21 | Manipur | 28 | Punjab | 35 | Uttar Pradesh |
| 36 | | | | West Bengal | | | | | |

Notes: There are 29 states and seven union territories in India; Rajasthan is the largest state in India in terms of area. Punjab and Haryana share the common capital "Chandigarh". Uttar Pradesh is the most populated state.

There are several factors behind the differences in the state-level impact of monetary policy and followed credit policy in each state. These include, among other factors, the following: firstly, state-wise differences in the mix of various industries; secondly, variations in the mix of large and small firms across states; and thirdly, the differences in financial deepening across states. It is well known that credit demand is different among different industries. This is attributed to the differences in working capital polices followed by firms. The fact is that there are some industries that are credit dependent such as the manufacturing industry as compared to service or agriculture ones. Thus, relatively industrialized states adopt different working capital policies.

Furthermore, monetary policy shocks have more impact on industrialized states than other less industrialized states. Another factor is that monetary policy affects the ability of banks to extend loans and advances to firms. Usually, due to an informant and transaction cost, small firms resort to dealing with banks and other relevant financial intermediaries to meet their need for operational working capital funds. On the contrary, large firms have a wider scope of acquiring funds from nonbanking sources. Therefore, states that have more small firms than large firms would be sensitive to the changes and adjustments in monetary policy instruments and would adopt relatively different working capital financing and investing policies.

Contemporary research work on credit policy channels suggests that the mix of small and large firms is an important determinant of monetary policy implementation. Kashyap and Stein [21] indicated that in regions that have quite large bank-dependent firms and a large number of small banks, monetary policy has more impact. Therefore, in regions/states with a low percentage of small banks and few bank-dependent customers, the credit policy will be weaker. In India, the process of financial deepening has not been uniform across states. Some states have experienced significant growth in banking and insurance activities, while some other states remain relatively underbanked. It might, therefore, be possible to envisage working capital policies across banked states that are more prone to the effects of a monetary policy shock as compared with those which are not [20]. The above discussion provides a ground for investigating the working capital management policies adopted by firms across all Indian states and to evaluate their impact on firms' performance.

Our motivation for this work is that working capital policies followed by Indian firms can play a substantial role in formulating liquidity, profitability, investment, and solvency policies and accordingly affect corporate financial sustainability. Our research evaluates the impact of working capital policies on firms' profitability with state-wise benchmarks. Hence, the proposed model of the present study warns companies to restructure their working capital policies to increase firms' profitability and ensure sustainability. The findings of the present study have added a new insight to a recent and important area which is corporate sustainability. Working capital management policies are not only important for firms' profitability, but they are also substantially significant for long- and short-term corporate sustainability, taking into consideration some economic and state-

wise factors. Sustainability has become a national and international concern; hence, the link between working capital policies and corporate sustainability is necessary for better performance of the Indian firms at the long- and short-term. The results of the study highlight that a conservative investment policy at the growth stage negatively influences firms' profitability, indicating aggressive policies could be preferred. The results reported that some companies did not considerably account for effective working capital policies while formulating working capital, which may significantly and negatively affect financial sustainability. The results indicated that conservative financing policy followed by Indian firms negatively affects firms' profitability measured by return on assets. This result warns companies to restructure their investment policy at the long run for better financial sustainability.

In this context, the study raises two questions:

What are the types of working capital financing and investment policies adopted by Indian firms across different states?

What is the impact of the adopted working capital policies on the financial performance of Indian firms located in different states?

To answer these questions, this study is going to find out the type of working capital financing and investment policies adopted by Indian firms across different states and examine the impact of the adopted working capital policies on the financial performance of Indian firms located in different states. To achieve the objectives of this research, the study extracted data for the period from 2011 to 2017 for 2181 manufacturing companies. After screening the data, some companies were excluded from the sample, and therefore, the final sample consists of 829 companies. A panel fixed- and random-effect model approach was adopted to estimate the results.

The remainder of this study is organized as follows: Section 2 presents the findings from previous literature on working capital policies; Section 3 provides an overview of Indian states and financial implications; Section 4 illustrates the research methodology; Section 5 demonstrates the analysis and discusses the findings; and Section 6 concludes the study.

## 2. Literature Review

Most of the empirical literature on working capital is based on developed countries such as the US, UK, Germany, Italy, Japan, etc., e.g., [4] in Belgium, Högerle et al [22] in Germany, [23] in Jordan, [24] in Spain, [25] in Norway, and [26] in the UK. Furthermore, there are some relevant issues that are not adequately addressed in the literature such as working capital policies. Nevertheless, in recent years, the issue of working capital management policies and its impact on firms' performance has gathered momentum in developing and emerging economies. Repeatedly, different business environments in emerging economies, including India, have highlighted the importance of working capital management policies and their impact on firm performance. Notwithstanding these, there is insufficient work on developing countries.

Research work regarding working capital policies revolves around two issues: working capital financing policy and investing policy. Viskari [27] noted that current literature has extensively and widely been focused on working capital efficiency, whereas working capital financing and investment policies have received considerable and thorough attention.

Afza and Nazir [28] demonstrated a negative correlation between the degree of aggressiveness of working capital financing and investing policy, and firms' profitability. Similarly, Al-Shubiri [17] also found out that the degree of aggressiveness of working capital negatively impacts firms' performance. The study utilized data for 59 companies for the period between 2004 and 2007. Bei and Wijewardana [10] evaluated the impact of working capital policies on 155 companies in Sri Lanka. The financial data used for analysis constituted a sample period of 5 years. The study suggested that working capital policy practitioners have to manage time constraints faced by many firms appropriately. Afza and Nazir [13], Al-Shubiri [17] and Hassani and Tavosi [29] used return on assets

(ROA) return on equity (ROE) and Tobin Q as dependent variables for measuring firms' financial performance. Panda and Nanda [30] aimed to empirically examine the relationship between working capital financing policy and the profitability of 1211 Indian firms from six manufacturing sectors, covering data for the period from 2000 to 2016. Results showed a convex relationship between working capital financing policy and profitability of construction, chemical, consumer, and goods firms.

Weinraub and Visscher [31] aimed to investigate the relationship between aggressive and conservative working capital policies and firms' performance sustainability of 10 industries. Results revealed that there is a significant negative association between industry assets and liabilities. Vahid et al. [18] examined the impact of working capital policies on the profitability of 28 Iranian companies. Results showed that conservative investment and aggressive financing policies have a negative effect on firms' performance. The authors of [8] investigated the impact of working capital management policy on customer electronic industry during the period from 1994 to 2004.

A recent study was conducted by Pestonji and Wichitsathian [19] to examine the effect of working capital investment and financing policies on firms' performances sustainability. The study took a sample of 68 listed firms. It was found that working capital investment policy has a positive and significant impact on firms' profitability, whereas working capital financing policy negatively impacts firms' profitability. Al-Shubiri [17] analyzed the impact of aggressive and conservative working capital management policies on firms' profitability of 59 listed at the Amman Stock exchange in Jordan. He found out that there is a negative relationship between aggressive investment policy and Tobin's Q. Furthermore, he found evidence that substantiated that aggressive financing policy positively impacts Tobin's Q of firms listed at the Amman Stock exchange. Vahid et al. [18] examined the relationship between aggressive and conservative working capital policies on the profitability of some firms listed on the Tehran Stock Exchange over the period from 2005 to 2009. Findings revealed that conservative investment policy negatively impacts firms' profitability, while aggressive investment policy positively impacts firms' profitability. In terms of financing policy, it was found that there is a negative association between aggressive financing policy and firm profitability, whereas the association between conservative financing policy and firm profitability is positive.

Another study conducted by Khajehpour et al. [32] investigated the relationship between aggressive working capital management policy and firms' profitability on a sample of 71 firms listed on the Tehran Stock Exchange. It was found out that aggressive working capital financing policy has an insignificant positive impact on firms' profitability measured by return on assets. On the contrary, working capital financing policy has a significant positive impact on Tobin's. In another context, Javid and Zita [33] analyzed the impact of working capital policy and profitability of 20 Pakistani firms listed on the Karachi Stock Exchange for the period from 2006 to 2011. Results showed that there is a significant negative association between working capital policies and firms' profitability. The authors of [10] evaluated working capital management policies in the Sri Lankan context for 155 firms listed in Colombo Stock Exchange for the period from 2002 to 2006. The study used multiple regression models for data analysis.

Arvind [34] conducted a study to evaluate the difficulties that manufacturing firms face to get access to finance. Three-thousand manufacturing firms across Indian states were surveyed in 2016. Chartered accountants, lawyers, and secretaries were also interviewed to provide details that are more relevant. The study showed that 46 percent of firms do not find any obstacle in getting finance from financial institutions, while 54 percent of companies face moderate to severe obstacles in getting finance. India has not paid enough attention to the centrality of business environment for wealth creation. As a result, firms and entrepreneurs face difficulties in doing business due to the bureaucracy, red-tape, and regulations. Liberalization and the institutional reforms that followed in 1991 created an enabling business environment that stimulated and promoted business operations. However, due to different legislations and the nature in which reforms were adopted

and implemented by states, some states did not considerably improve the ease of doing business quicker as compared to others. Working capital policies depend on the business environment, legislation, and state fiscal policies.

Matteo et al. [35] attempted to find out which financing method is suitable for small and medium enterprises by comparing firms that are working in Italy and Germany. Results revealed that there are many differences between German and Italian small- and medium-sized firms regarding the appropriate financing strategy. The degree of usefulness of alternative debt-funding instruments continues to be affected by the examined space-time background. In the case of Italy, the usefulness of these instruments is relatively low. In the case of Germany, the situation is the polar opposite. Giacosa et al. [36] aimed to evaluate the effect of firms specifically on the financial leverage of small- and medium-sized firms. The study targeted all Italian manufacturing companies, the final sample consisted of 4705 companies. Findings showed relatively moderated association between fixed assets and liabilities. Furthermore, it was found that fixed assets were not funded by permanent capital which created financial tension. Moreover, it was revealed that there is a strong relationship association between revenues and liabilities. Rossi et al. [37] tried to evaluate the effect of firms specific on the financial choice of small- and medium-sized firms. Results revealed that access to finance is very important for small and medium firms as various firms have different needs and encounter various challenges in accessing external finance in comparison to large firms. Furthermore, it was found that due to the low level of investment in equity, small firms are more dependent on other external sources of finance.

Based on the above argument, it can be said that although the theoretical and empirical literature on working capital management policies is fairly well researched, it is far from complete, and there are still many issues that need to be addressed and thoroughly researched. This paper extends the existing working capital policies literature in several dimensions. Firstly, this paper attempts to fill an important gap in the existing literature by providing rigorous econometric evidence on the impact of working capital management policies on firms' profitability in India, using new panel data set based on 829 companies for the period from 2011 to 2017. Secondly, this work is based on a large sample from different states, which enables researchers to come up with an appropriate scientific generalization. Thirdly, applying a panel data model which avoids the weakness of ordinary least square regression to the data of listed manufacturing companies working in Indian states overcomes the methodological issues associated with previous studies and produce results which enriches the literature of working capital management policies.

## 3. Methodology

### 3.1. Data and Sample Selection

The initial sample consists of 2181 manufacturing companies drawn from Prowess IQ database. Financial firms and other nonmanufacturing firms were not included in the initial sample because they have a different capital structure. The following criterion is set for any company to be included in the sample:

- A company must belong to the manufacturing sector.
- A company must have data for the study period from March 2011 to March 2017 financial year.
- A company must belong to a state that is represented by enough number of companies for running the analysis.

Therefore, firms that did not have data for the study period were dropped from the sample. There are 29 states and 7 union territories in India. Out of them, only 17 states were included in this analysis, and the remaining ones were excluded for two reasons. Firstly, they were not represented by any listed company or were represented by insufficient companies, e.g., (1, 2, 3, 4, or 5) which are not enough for running the econometric model. When the number of cross-sections is not more than the number of repressors, which is five in this study, the analysis cannot be done. Subsequently, some companies were excluded

from the sample due to them belonging to states that were not represented enough to be analyzed. Therefore, the final sample consists of 829 companies from 17 states as shown in Table 2. The study employed data for the period from March 2011 to March 2017.

**Table 2.** Selected states and their number of represented companies.

| State | NO Firms | NO Observations | State | NO Firms | NO Observations | State | NO Firms | NO Observations |
|---|---|---|---|---|---|---|---|---|
| Andhra Pradesh | 14 | 119 | Kerala | 6 | 42 | Rajasthan | 30 | 210 |
| Dadra & Nagar Haveli | 11 | 77 | Madhya Pradesh | 25 | 175 | Tamil Nadu | 76 | 532 |
| Gujarat | 102 | 714 | Maharashtra | 232 | 1624 | Telangana | 61 | 427 |
| Haryana | 35 | 245 | NCT of Delhi | 73 | 511 | Uttar Pradesh | 32 | 224 |
| Himachal Pradesh | 10 | 70 | Odisha | 12 | 84 | West Bengal | 51 | 357 |
| Karnataka | 33 | 231 | Punjab | 26 | 182 | Total | 829 | 5803 |

The study uses probability sampling. A stratified random sampling was used. Manufacturing firms were classified on the basis of their location and the state that they are operating in. This method is acknowledged by different researchers [38]. The present study has used this method in order to classify the manufacturing firms on the basis of their location and to allow fair representation of different strata in the final sample.

*3.2. Study Variables*

3.2.1. Dependent Variables

It is argued that market-based measures are more likely to be suitable than accounting-based measures. It is also believed that several uncontrollable factors affect them [39]. For reflecting the results of management actions, Hutchinson and Gul [40] argued that accounting-based measures are preferable over market-based measures. In line with previous research, the study takes ROA, for measuring firms' profitability (e.g., [16,28–30,41–43]).

3.2.2. Independent Variables

The study focuses on two working capital policies: working capital investing policy and financing policy. These two constitute the independent variables of the study.

- Working capital financing policy

When a company finances its current assets and part of its fixed assets from short-term sources of funds, it entails that the company is following an aggressive financing policy. By contrast, when a company finances its fixed assets and part of its current assets from long-term capital sources, it is held that the company is following a conservative financing policy [28]. The aggressiveness of financing policy can be measured as follows:

$$FP = \text{Current liabilities/total assets}$$

High ratio, i.e., more than 50%, implies that firms are following relatively aggressive financing policy, while low ratio, i.e., less than 50%, implies that firms are following relatively conservative financing policy [44].

- Working capital investing policy

If a firm follows an aggressive investment policy, which is a situation where a company invests less in current assets and more in fixed assets, the opposite takes place when the company follows a conservative policy [28]. For measuring the aggressiveness of the investment policy, the following formula is applied:

$$IP = \text{current assets/total assets}$$

High ratio, i.e., more than 50%, implies that firms are following relatively conservative investment policy, while low ratio, i.e., more than 50%, implies that firms are following relatively aggressive investment policy [44].

### 3.2.3. Control Variables

The study control variables are drawn from prior research on working capital management, e.g., [4,8,26,28,45–49]. The following control variables are used:

- Leverage
- Firm's size
- Firm's age

In all regressions, the same set of control variables is used. Figure 1 shows the research framework of the study.

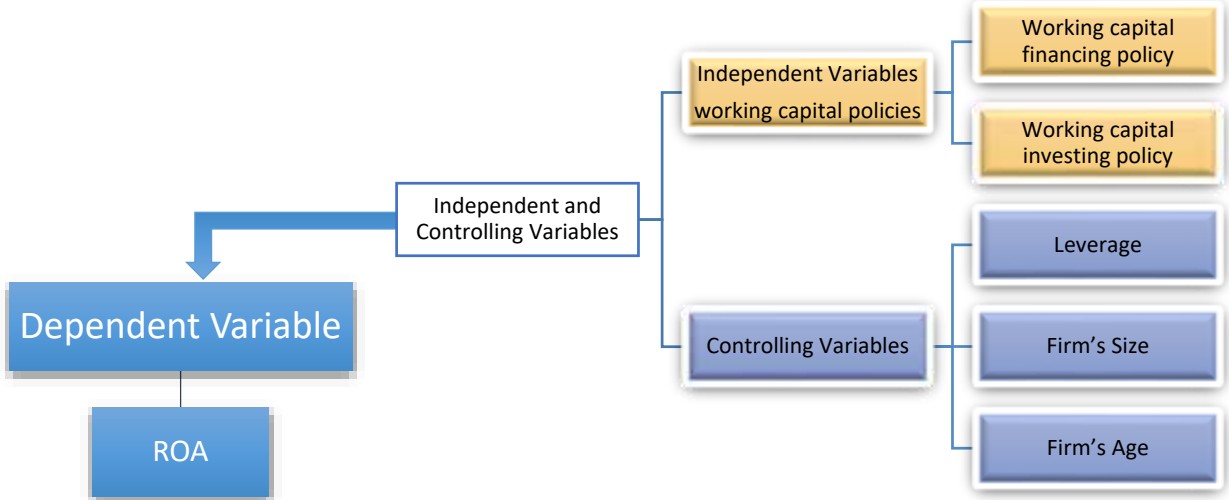

**Figure 1.** Research framework.

### 3.3. Hypotheses and Models Development

To examine the impact of working capital financing policy on firms' profitability, the hypothesis is formulated as follows:

**Hypothesis 1 (H1).** *Working capital financing policy negatively and significantly impacts firms' profitability.*

To investigate the effect of working capital investment policy on firms' profitability, the study formulated the following hypothesis:

**Hypothesis 2 (H2).** *Working capital investment policy significantly and negatively impacts firms' profitability.*

Subsequently, the study model is developed for testing the hypothesis as follows.

$$\text{ROA}_{it} = a_0 + a_1\,\text{FP}_{it} + a_2\,\text{IP}_{it} + a_3\,\text{LEVE}_{it} + a_4\text{size}_{it} + a_5\text{AGE}_{it} + a_6\varepsilon_{it}$$

where $\alpha$ is the intercept, $\varepsilon$ is the error term of the model, i and t correspond to firm and year, ROA is return on assets, IP is working capital investment policy, FP is working capital financing policy, size is firm size that is measured by the natural log of total assets, LEVE stands for leverage, and AGE is firm age, which is calculated by the number of years in which the company is operating.

## 4. Results of the Study

- Descriptive statistics

The first step of any analysis starts with descriptive statistics, which describes the sample characteristics. It shows the central tendency measures like mean, standard deviation, variation, frequencies, and minimum and maximum values. The present study shows the main and standard deviation of the selected sample over the study period. Mean is the arithmetical average of the variable, while standard deviation refers to the variation in the data.

The results in Table 3 demonstrate the descriptive statistics for the study variables return on assets, financing policy, investing policy, leverage, and firms' size and age. The ROA mean value of all companies across all states covered in this study ranges between 1 and 4.31 percent. The maximum ROA is 4.10, which is achieved by a firm located in Maharashtra state. The financing policy mean values for firms located in different states are ranging between 0.34 and 0.44, which indicates that companies working in different states adopt a conservative financing policy. The whole-sample descriptive statistics confirm that Indian manufacturing companies adopt a conservative financing policy. The mean value of the financing policy ratio is 0.38. The extreme conservative financing policy is found to be adopted by some companies located in Maharashtra, Telangana, NCT of Delhi, West Bengal, Karnataka, and Madhya Pradesh states. The minimum values of the financing policy are 0.00, 0.00, 0.00, 0.01, 0.02, and 0.02. This means that these companies utilize other sources of finance such as issuing ordinary shares, preferred shares, long-term debt, and retained earnings. Regarding investing policy, the mean values of investment policy ratio for all manufacturing companies located in different states range between 0.43 and 0.61. Furthermore, the mean value of the investment policy ratio for the whole sample is 0.51. Moreover, results reveal that companies that are carrying out their business in Madhya Pradesh, Dadra and Nagar, Gujarat, Maharashtra, Rajasthan, NCT of Delhi and Uttar Pradesh states are adopting a conservative investing policy. The mean values of the investing policy ratio are 0.61, 0.54, 0.53, 0.52, 0.51, 0.52, and 0.52 respectively. This indicates that these companies are investing more in current assets than fixed assets. The extreme conservative investing policy is found to be followed by manufacturing companies working in Maharashtra and NCT of Delhi, the maximum values of the investing policy ratio in these states are 0.97 and 0.96. A higher ratio means a more conservative investment policy [31]. The leverage mean values of manufacturing companies doing business in Andhra Pradesh, Kerala, Himachal Pradesh, Punjab, and Tamil Nadu are 2.05, 2.14. 2.53, 2.78, and 5.22, respectively, while the leverage of the manufacturing companies that are doing business in the other states have the mean below 2.

**Table 3.** State-wise descriptive statistics.

| State | Variable | Minimum | Maximum | Mean | Std. Deviation | State | Variable | Minimum | Maximum | Mean | Std. Deviation |
|---|---|---|---|---|---|---|---|---|---|---|---|
| Andhra Pradesh | ROA | −35.89 | 29.34 | 4.31 | 9.90 | NCT of Delhi | ROA | −20.38 | 28.78 | 3.11 | 4.63 |
| | FP | 0.08 | 0.78 | 0.35 | 0.14 | | FP | 0.00 | 0.83 | 0.39 | 0.17 |
| | IP | 0.08 | 0.89 | 0.49 | 0.20 | | IP | 0.05 | 0.96 | 0.52 | 0.19 |
| | LEV | 0.01 | 32.58 | 2.05 | 4.60 | | LEV | 0.00 | 29.54 | 1.19 | 1.77 |
| | SIZE | 2.21 | 4.56 | 3.44 | 0.58 | | SIZE | 1.26 | 6.06 | 3.41 | 0.83 |
| | AGE | 18.00 | 70.00 | 32.00 | 13.45 | | AGE | 6.00 | 128.00 | 29.37 | 14.60 |
| Dadra & Nagar Haveli | ROA | −12.90 | 10.82 | 2.47 | 3.88 | Odisha | ROA | −9.30 | 10.94 | 2.18 | 4.11 |
| | FP | 0.18 | 0.71 | 0.44 | 0.12 | | FP | 0.15 | 0.70 | 0.34 | 0.12 |
| | IP | 0.18 | 0.91 | 0.54 | 0.18 | | IP | 0.09 | 0.85 | 0.46 | 0.19 |
| | LEV | 0.05 | 9.25 | 1.50 | 1.20 | | LEV | 0.20 | 11.90 | 1.45 | 2.31 |
| | SIZE | 2.41 | 5.57 | 3.52 | 0.82 | | SIZE | 2.00 | 4.77 | 3.44 | 0.88 |
| | AGE | 14.00 | 57.00 | 30.45 | 11.97 | | AGE | 20.00 | 70.00 | 41.83 | 16.67 |
| Gujarat | ROA | −35.07 | 27.64 | 4.09 | 5.87 | Punjab | ROA | −18.04 | 23.99 | 1.70 | 5.87 |
| | FP | 0.06 | 0.89 | 0.38 | 0.15 | | FP | 0.06 | 0.72 | 0.37 | 0.15 |
| | IP | 0.14 | 0.95 | 0.53 | 0.17 | | IP | 0.09 | 0.90 | 0.46 | 0.20 |
| | LEV | 0.00 | 120.49 | 1.46 | 4.65 | | LEV | 0.02 | 133.00 | 2.78 | 10.15 |
| | SIZE | 1.68 | 5.32 | 3.37 | 0.83 | | SIZE | 2.12 | 4.89 | 3.62 | 0.70 |
| | AGE | 12.00 | 98.00 | 30.73 | 14.70 | | AGE | 11.00 | 74.00 | 32.88 | 13.40 |
| Haryana | ROA | −29.77 | 37.09 | 2.77 | 5.76 | Rajasthan | ROA | −12.49 | 24.64 | 4.29 | 5.31 |
| | FP | 0.07 | 0.77 | 0.42 | 0.17 | | FP | 0.08 | 0.75 | 0.36 | 0.14 |
| | IP | 0.06 | 0.88 | 0.48 | 0.20 | | IP | 0.19 | 0.96 | 0.51 | 0.16 |
| | LEV | 0.01 | 64.17 | 1.99 | 5.51 | | LEV | 0.03 | 4.52 | 1.51 | 1.11 |
| | SIZE | 1.53 | 5.83 | 3.47 | 0.86 | | SIZE | 2.20 | 5.05 | 3.50 | 0.72 |
| | AGE | 17.00 | 115.00 | 34.94 | 18.22 | | AGE | 6.00 | 79.00 | 33.50 | 15.02 |
| Himachal Pradesh | ROA | −44.62 | 14.22 | 1.00 | 7.51 | Tamil Nadu | ROA | −26.98 | 22.13 | 2.32 | 5.99 |
| | FP | 0.12 | 0.61 | 0.37 | 0.11 | | FP | 0.03 | 0.91 | 0.40 | 0.15 |
| | IP | 0.12 | 0.79 | 0.43 | 0.16 | | IP | 0.10 | 0.92 | 0.47 | 0.17 |
| | LEV | 0.02 | 43.21 | 2.53 | 5.54 | | LEV | 0.00 | 664.75 | 5.22 | 34.88 |
| | SIZE | 2.75 | 3.89 | 3.30 | 0.33 | | SIZE | 1.84 | 5.19 | 3.57 | 0.68 |
| | AGE | 7.00 | 37.00 | 28.40 | 7.18 | | AGE | 5.00 | 107.00 | 37.53 | 19.08 |

**Table 3.** *Cont.*

| State | Variable | Minimum | Maximum | Mean | Std. Deviation | State | Variable | Minimum | Maximum | Mean | Std. Deviation |
|---|---|---|---|---|---|---|---|---|---|---|---|
| Karnataka | ROA | −39.73 | 26.42 | 3.01 | 8.01 | Telangana | ROA | −34.67 | 34.23 | 3.08 | 6.75 |
| | FP | 0.02 | 0.74 | 0.36 | 0.16 | | FP | 0.00 | 0.96 | 0.36 | 0.17 |
| | IP | 0.08 | 0.81 | 0.45 | 0.18 | | IP | 0.00 | 0.90 | 0.50 | 0.18 |
| | LEV | 0.01 | 39.68 | 1.21 | 2.81 | | LEV | 0.00 | 68.98 | 1.57 | 4.73 |
| | SIZE | 1.99 | 5.63 | 3.72 | 0.80 | | SIZE | 1.31 | 5.28 | 3.29 | 0.77 |
| | AGE | 12.00 | 74.00 | 32.91 | 14.70 | | AGE | 4.00 | 71.00 | 28.08 | 10.83 |
| Kerala | ROA | −9.58 | 28.33 | 3.68 | 8.63 | Uttar Pradesh | ROA | −27.28 | 32.29 | 3.61 | 5.52 |
| | FP | 0.10 | 0.82 | 0.37 | 0.20 | | FP | 0.07 | 0.71 | 0.38 | 0.14 |
| | IP | 0.10 | 0.83 | 0.50 | 0.22 | | IP | 0.19 | 0.92 | 0.52 | 0.17 |
| | LEV | 0.02 | 6.74 | 2.14 | 2.06 | | LEV | 0.06 | 9.01 | 1.31 | 1.26 |
| | SIZE | 2.04 | 3.84 | 3.12 | 0.52 | | SIZE | 1.95 | 5.77 | 3.66 | 0.89 |
| | AGE | 6.00 | 54.00 | 27.00 | 13.58 | | AGE | 11.00 | 86.00 | 35.09 | 16.08 |
| Madhya Pradesh | ROA | −54.43 | 14.71 | 2.47 | 6.14 | West Bengal | ROA | −35.07 | 30.94 | 2.38 | 6.81 |
| | FP | 0.02 | 1.31 | 0.42 | 0.17 | | FP | 0.01 | 0.83 | 0.37 | 0.17 |
| | IP | 0.24 | 0.86 | 0.61 | 0.16 | | IP | 0.02 | 0.90 | 0.48 | 0.18 |
| | LEV | 0.05 | 57.19 | 1.69 | 5.18 | | LEV | 0.00 | 58.75 | 1.73 | 4.05 |
| | SIZE | 1.68 | 5.31 | 3.03 | 0.74 | | SIZE | 1.68 | 5.74 | 3.65 | 0.75 |
| | AGE | 12.00 | 72.00 | 29.32 | 13.55 | | AGE | 7.00 | 117.00 | 47.88 | 27.28 |
| Maharashtra | ROA | −50.87 | 47.10 | 3.56 | 6.86 | All states | ROA | −54.43 | 47.10 | 3.19 | 6.39 |
| | FP | 0.00 | 0.88 | 0.37 | 0.16 | | FP | 0.00 | 1.31 | 0.38 | 0.16 |
| | IP | 0.00 | 0.97 | 0.52 | 0.18 | | IP | 0.00 | 0.97 | 0.51 | 0.18 |
| | LEV | 0.00 | 249.54 | 1.59 | 7.94 | | LEV | 0.00 | 664.75 | 1.93 | 11.93 |
| | SIZE | 1.56 | 6.74 | 3.61 | 0.88 | | SIZE | 1.53 | 6.74 | 3.50 | 0.83 |
| | AGE | 3.00 | 154.00 | 40.56 | 23.67 | | AGE | 3.00 | 154.00 | 35.53 | 19.58 |

- Correlation matrix and multicollinearity test

Pearson correlation matrix was performed to find out the association between working capital financing, investment policy, and firms' performance. Pearson correlation matrix demonstrates the type of association that exists and whether it is significant or not. It also gives a signal if multicollinearity exists. Table 4 reveals that there is no multicollinearity problem because there is no high correlation between the independent variables. For making sure regarding the absence of multicollinearity, VIF values were calculated. Results in Table 4 confirm the absence of multicollinearity because the heist VIF value for independent variables is 5.42. Statisticians believe that multicollinearity exists if VIF is equal to or greater than 10.

Findings in Table 4 reveal that all independent variables have a significant association with firms' performance measured by ROA. Results show that the financing policy ratio has a substantial and negative association with return on assets of firms located in all states included in this study except for companies located in Himachal Pradesh that have an insignificant negative association.

Regarding investing policy, the findings in Table 4 show that there is a significant and positive correlation between return on assets and the working capital financing policy of firms that are doing business in Andhra Pradesh, Karnataka, Kerala, Maharashtra, Punjab, Telangana, and West Bengal states. By contrast, the working capital investing policy significantly and negatively correlates with the profitability of companies working in Dadra and Nagar Haveli union territories and insignificantly correlates with the profitability of companies that are doing business in Gujarat, Haryana, NCT of Delhi, Odisha, and Rajasthan states. Leverage ratio has a negative and significant relationship with the profitability of firms that are carrying out business in all Indian states except Odisha and Himachal Pradesh, in which there is an insignificant negative association between firms' profitability and leverage ratio. Firms' size positively associates with return on assets ratio of all companies operating in all Indian states except Dadra and Nagar Haveli, Himachal Pradesh, and Uttar Pradesh, in which firm's size associates negatively and significantly with firms' performance. This could be attributed to the fact that large firms have the advantage of exploiting any available investment opportunity. Moreover, they have more funds to invest and have the ability to raise funds from external sources when needed. Furthermore, companies' age correlates negatively and significantly with the profitability of firms that are working in Dadra and Nagar Haveli, NCT of Delhi, and Uttar Pradesh states. However, it correlates positively and significantly with the profitability of firms' that are working in Himachal Pradesh, Kerala, Telangana, and West Bengal.

- Impact of working capital policies on firms' profitability

Multiple regression is a statistical tool used for finding out the impact of the independent variable over the dependent variable. It specifies whether the impact is significant or not. It also reports the portion of changes in the dependent variable corresponding to the change in the independent variable. In the present study, multiple regression models have been used to examine the impact of working capital policies on firms' profitability. For obtaining accurate and reliable results, all assumptions of regression models have to be met, which has been done in the present study. The scatterplot technique was used for examining normality and linearity. VIF was used for detecting multicollinearity, as shown in Table 4.

Moreover, heteroscedastic was checked and talked where existFurthermore, autocorrelation assumption was also verified by using the Durbin–Waston test and LM test where run. After fulfilling all the assumptions, the study conducted a redundant fixed-effect test to decide which model is appropriate for the study (pooled or fixed and random effect) and whether it is a model with one-way intercept or two-way intercept.

**Table 4.** State-wise correlation matrix.

| State | | ROA | FP | IP | LEV | SIZE | AGE | State | | ROA | FP | IP | LEV | SIZE | AGE |
|---|---|---|---|---|---|---|---|---|---|---|---|---|---|---|---|
| | ROA | 1 | | | | | | | ROA | 1 | | | | | |
| | FP | −0.205 * | 1.00 | | | | | | FP | −0.013 | 1 | | | | |
| | IP | 0.536 ** | 0.233 * | 1.00 | | | | | IP | −0.027 | 0.450 ** | 1 | | | |
| Andhra Pradesh | LEV | −0.340 ** | 0.251 * | −0.361 ** | 1.00 | | | NCT of Delhi | LEV | −0.099 * | 0.369 ** | 0.126 ** | 1 | | |
| | SIZE | 0.342 ** | −0.402 ** | −0.344 ** | −0.09 | 1.00 | | | SIZE | 0.254 ** | 0.235 ** | −0.093 * | 0.061 | 1 | |
| | AGE | −0.111 | −0.11 | −0.18 | 0.02 | 0.278 ** | 1.00 | | AGE | −0.123 ** | 0.106 * | 0.138 ** | 0.078 | 0.185 ** | 1 |
| | VIF | | 1.34 | 1.47 | 1.36 | 1.39 | 1.09 | | VIF | | 1.58 | 1.36 | 1.17 | 1.16 | 1.07 |
| | ROA | 1 | | | | | | | ROA | 1 | | | | | |
| | FP | −0.403 ** | 1.00 | | | | | | FP | −0.326 ** | 1 | | | | |
| | IP | −0.241 * | 0.827 ** | 1.00 | | | | | IP | −0.109 | 0.7000** | 1 | | | |
| Dadra & Nagar Haveli | LEV | −0.497 ** | 0.12 | −0.15 | 1.00 | | | Odisha | LEV | −0.056 | 0.249 * | .177 | 1 | | |
| | SIZE | −0.126 | −0.13 | −0.256 * | 0.624 ** | 1.00 | | | SIZE | 0.301 ** | −0.630 ** | −0.637 ** | −0.238 * | 1 | |
| | AGE | −0.292 ** | 0.657 ** | 0.535 ** | −0.01 | −0.07 | 1.00 | | AGE | 0.180 | −0.289 ** | −0.318 ** | −0.134 | 0.792 ** | 1 |
| | VIF | | 5.24 | 4.00 | 2.12 | 1.79 | 1.83 | | VIF | | 2.47 | 2.37 | 1.08 | 6.00 | 3.53 |
| | ROA | 1 | | | | | | | ROA | 1 | | | | | |
| | FP | −0.299 ** | 1.00 | | | | | | FP | −0.266 ** | 1 | | | | |
| | IP | −0.002 | 0.573 ** | 1.00 | | | | | IP | 0.160 * | 0.140 | 1 | | | |
| Gujarat | LEV | −0.355 ** | 0.154 ** | 0.04 | 1.00 | | | Punjab | LEV | −0.247 ** | 0.178 * | −0.115 | 1 | | |
| | SIZE | 0.064 | −0.232 ** | −0.418 ** | 0.099 ** | 1.00 | | | SIZE | 0.037 | 0.046 | −0.349 ** | 0.120 | 1 | |
| | AGE | −0.001 | −0.221 ** | −0.235 ** | −0.06 | 0.204 ** | 1.00 | | AGE | −0.089 | 0.018 | 0.039 | 0.204 ** | −0.083 | 1 |
| | VIF | | 1.55 | 1.72 | 1.05 | 1.25 | 1.09 | | VIF | | 1.07 | 1.19 | 1.11 | 1.17 | 1.06 |

**Table 4.** *Cont.*

| State | | ROA | FP | IP | LEV | SIZE | AGE | State | | ROA | FP | IP | LEV | SIZE | AGE |
|---|---|---|---|---|---|---|---|---|---|---|---|---|---|---|---|
| Haryana | ROA | 1 | | | | | | Rajasthan | ROA | 1 | | | | | |
| | FP | −0.238 ** | 1.00 | | | | | | FP | −0.336 ** | 1 | | | | |
| | IP | −0.053 | 0.767 ** | 1.00 | | | | | IP | −0.036 | 0.612 ** | 1 | | | |
| | LEV | −0.253 ** | 0.08 | −0.02 | 1.00 | | | | LEV | −0.354 ** | 0.423 ** | −0.056 | 1 | | |
| | SIZE | 0.051 | 0.02 | −0.161 * | 0.199 ** | 1.00 | | | SIZE | 0.302 ** | −0.134 | −0.443 ** | 0.096 | 1 | |
| | AGE | −0.036 | 0.350 ** | 0.257 ** | −0.01 | 0.319 ** | 1.00 | | AGE | 0.015 | −0.048 | −0.349 ** | 0.115 | 0.491 ** | 1 |
| | VIF | | 2.69 | 2.66 | 1.07 | 1.28 | 1.31 | | VIF | | 2.54 | 2.57 | 1.51 | 1.50 | 1.39 |
| Himachal Pradesh | ROA | 1 | | | | | | Tamil Nadu | ROA | 1 | | | | | |
| | FP | −0.174 | 1.00 | | | | | | FP | −0.241 ** | 1 | | | | |
| | IP | 0.137 | 0.589 ** | 1.00 | | | | | IP | 0.080 | 0.549 ** | 1 | | | |
| | LEV | −0.150 | 0.23 | 0.438 ** | 1.00 | | | | LEV | −0.204 ** | 0.078 | −0.047 | 1 | | |
| | SIZE | −0.028 | −0.11 | −0.13 | 0.12 | 1.00 | | | SIZE | 0.257 ** | −0.176 ** | −0.388 ** | −0.057 | 1 | |
| | AGE | 0.399 ** | −0.05 | 0.16 | 0.02 | 0.13 | 1.00 | | AGE | −0.035 | −0.244 ** | −0.221 ** | 0.070 | 0.361 ** | 1 |
| | VIF | | 1.58 | 1.98 | 1.30 | 1.09 | 1.10 | | VIF | | 1.51 | 1.67 | 1.04 | 1.33 | 1.21 |
| Karnataka | ROA | 1 | | | | | | Telangana | ROA | 1 | | | | | |
| | FP | −0.318 ** | 1.00 | | | | | | FP | −0.152 ** | 1 | | | | |
| | IP | 0.243 ** | 0.369 ** | 1.00 | | | | | IP | 0.166 ** | 0.465 ** | 1 | | | |
| | LEV | −0.220 ** | 0.352 ** | −0.02 | 1.00 | | | | LEV | −0.160 ** | 0.273 ** | 0.044 | 1 | | |
| | SIZE | 0.060 | 0.09 | 0.131 * | 0.03 | 1.00 | | | SIZE | 0.348 ** | 0.028 | −0.088 | −0.049 | 1 | |
| | AGE | −0.035 | −0.06 | 0.04 | 0.200 ** | 0.03 | 1.00 | | AGE | 0.117 * | −0.111 * | 0.024 | −0.136 ** | 0.239 ** | 1 |
| | VIF | | 1.40 | 1.22 | 1.26 | 1.02 | 1.08 | | VIF | | 1.42 | 1.32 | 1.10 | 1.09 | 1.10 |

**Table 4.** *Cont.*

| State | | ROA | FP | IP | LEV | SIZE | AGE | State | | ROA | FP | IP | LEV | SIZE | AGE |
|---|---|---|---|---|---|---|---|---|---|---|---|---|---|---|---|
| | ROA | 1 | | | | | | | ROA | 1 | | | | | |
| | FP | −0.203 | 1.00 | | | | | | FP | −0.318 ** | 1 | | | | |
| | IP | 0.341 * | 0.530 ** | 1.00 | | | | | IP | 0.001 | 0.709 ** | 1 | | | |
| Kerala | LEV | −0.431 ** | 0.07 | −0.568 ** | 1.00 | | | Uttar Pradesh | LEV | −0.358 ** | 0.258 ** | −0.137 * | 1 | | |
| | SIZE | 0.217 | −0.461 ** | −0.380 * | −0.348 * | 1.00 | | | SIZE | −0.123 | −0.267 ** | −0.512 ** | 0.166 * | 1 | |
| | AGE | 0.333 * | −0.336 * | 0.22 | −0.460 ** | 0.16 | 1.00 | | AGE | −0.144 * | −0.080 | −0.191 ** | −0.025 | 0.125 | 1 |
| | VIF | | 2.64 | 4.26 | 5.42 | 2.40 | 1.54 | | VIF | | 2.79 | 3.24 | 1.39 | 1.39 | 1.06 |
| | ROA | 1 | | | | | | | ROA | 1 | | | | | |
| | FP | −0.360 ** | 1.00 | | | | | | FP | −0.314 ** | 1 | | | | |
| | IP | 0.029 | 0.455 ** | 1.00 | | | | | IP | 0.153 ** | 0.571 ** | 1 | | | |
| Madhya Pradesh | LEV | −0.045 | 0.07 | −0.14 | 1.00 | | | West Bengal | LEV | −0.295 ** | 0.191 ** | 0.012 | 1 | | |
| | SIZE | 0.149 * | 0.05 | −0.434 ** | 0.10 | 1.00 | | | SIZE | 0.298 ** | −0.166 ** | −0.141 ** | 0.160 ** | 1 | |
| | AGE | −0.005 | −0.158 * | −0.218 ** | −0.07 | 0.503 ** | 1.00 | | AGE | 0.105 * | −0.024 | 0.069 | 0.090 | 0.289 ** | 1 |
| | VIF | | 1.50 | 1.78 | 1.06 | 1.84 | 1.44 | | VIF | | 1.60 | 1.53 | 1.10 | 1.16 | 1.11 |
| | ROA | 1 | | | | | | | ROA | 1 | | | | | |
| | FP | −0.275 ** | 1.00 | | | | | | FP | −0.251 ** | 1 | | | | |
| | IP | 0.110 ** | 0.491 ** | 1.00 | | | | | IP | 0.103 ** | 0.507 ** | 1 | | | |
| Maharashtra | LEV | −0.159 ** | 0.112 ** | 0.02 | 1.00 | | | all states | LEV | −0.137 ** | 0.085 ** | −0.017 | 1 | | |
| | SIZE | 0.133 ** | −0.160 ** | −0.338 ** | −0.04 | 1.00 | | | SIZE | 0.113 ** | −0.14 ** | −0.27 ** | 0.007 | 1 | |
| | AGE | 0.001 | −0.129 ** | −0.152 ** | −0.04 | 0.373 ** | 1.00 | | AGE | −0.003 | −0.08 ** | −0.08 ** | 0.017 | 0.31 ** | 1 |
| | VIF | | 1.34 | 1.45 | 1.02 | 1.28 | 1.17 | | VIF | | 1.43 | 1.49 | 1.02 | 1.19 | 1.12 |

ROA stands for return on assets, FP stands for financing policy, IP stands for investing policy, LEV stands for leverage, SIZE stands for company size, AGE stands for age of the company, VIF stands for variance inflation factor, * means the correlation is significant at 0.05 level, ** means the correlation is significant at 0.01.

Table 5 shows that most of the models have one-way variable intercept whether it is cross-section fixed effects or period fixed effects, besides Madhya Pradesh, Maharashtra, and Rajasthan who had their data analyzed with two-ways intercept. Furthermore, the results of redundant fixed-effects tests reveal that a pooled regression model is appropriate for Himachal Pradesh's data analysis because its data is heterogeneous in nature. After checking the heterogeneity of data for all states, the Hausman Test was conducted to decide between fixed and random effect, as shown in Table 6. Results reveal that the fixed-effect model is preferred to random effect model in all cases except the cases of Dadra and Nagar Haveli, Karnataka, Odisha, Uttar Pradesh, and West Bengal.

**Table 5.** Redundant fixed-effects tests.

| State | Test Cross-Section Fixed Effects | | Test Period Fixed Effects | | State | Test Cross-Section Fixed Effects | | Test Period Fixed Effects | |
|---|---|---|---|---|---|---|---|---|---|
| | Chi-Square | Prob. | Chi-Square | Prob. | | Chi-Square | Prob. | Chi-Square | Prob. |
| Andhra Pradesh | 65.13 | 0.00 | 5.50 | 0.48 | NCT of Delhi | 341.45 | 0.00 | 3.82 | 0.58 |
| Dadra & Nagar Haveli | 55.93 | 0.00 | 2.52 | 0.87 | Odisha | 73.32 | 0.00 | 7.53 | 0.27 |
| Gujarat | 508.93 | 0.00 | 2.46 | 0.78 | Punjab | 115.96 | 0.00 | 11.04 | 0.05 |
| Haryana | 175.24 | 0.00 | 7.86 | 0.25 | Rajasthan | 74.94 | 0.00 | 15.58 | 0.01 |
| Himachal Pradesh | 15.69 | 0.07 | 10.27 | 0.11 | Tamil Nadu | 296.20 | 0.00 | 12.32 | 0.06 |
| Karnataka | 188.74 | 0.00 | 8.71 | 0.19 | Telangana | 233.96 | 0.00 | 11.75 | 0.07 |
| Kerala | 43.51 | 0.00 | 3.92 | 0.69 | Uttar Pradesh | 123.33 | 0.00 | 8.79 | 0.19 |
| Madhya Pradesh | 80.76 | 0.00 | 13.99 | 0.03 | West Bengal | 39.97 | 0.84 | 11.52 | 0.04 |
| Maharashtra | 1167.48 | 0.00 | 48.66 | 0.00 | all states | 2448.88 | 0.00 | 0.73 | 0.98 |

**Table 6.** Correlated random effects—Hausman test.

| State | Prob. | Fixed/Random | State | Prob | Fixed/Random | State | Prob. | Fixed/Random |
|---|---|---|---|---|---|---|---|---|
| Andhra Pradesh | 0.02 | Fixed | Kerala | 0.00 | Fixed | Rajasthan | 0.00 | Fixed |
| Dadra & Nagar Haveli | 0.51 | Random | Madhya Pradesh | 0.00 | Fixed | Tamil Nadu | 0.00 | Fixed |
| Gujarat | 0.00 | Fixed | Maharashtra | 0.00 | Fixed | Telangana | 0.00 | Fixed |
| Haryana | 0.01 | Fixed | NCT of Delhi | 0.03 | Fixed | Uttar Pradesh | 0.05 | Random |
| Himachal Pradesh | NA | Pooled | Odisha | 0.10 | Random | West Bengal | 0.36 | Random |
| Karnataka | 0.06 | Random | Punjab | 0.00 | Fixed | all states | 0.00 | Fixed |

In order to choose the most appropriate model (fixed or random model), the Hausman test was conducted, and according to the result of the test, the appropriate regression models are selected, and the results are shown in Table 7.

**Table 7.** Regression modes.

| | Fixed Effect | | | | Fixed Effect | | | | Fixed Effect | | |
|---|---|---|---|---|---|---|---|---|---|---|---|
| | Variable | Coefficient | Prob. | | Variable | Coefficient | Prob. | | Variable | Coefficient | Prob. |
| | FP | −20.35 | 0.00 | | FP | −12.99 | 0.22 | | FP | −13.32 | 0.00 |
| | IP | 45.14 | 0.00 | | IP | 28.64 | 0.08 | | IP | 18.84 | 0.00 |
| | LEVE | 0.39 | 0.01 | | LEVE | −0.06 | 0.96 | | LEVE | −0.01 | 0.36 |
| Andhra Pradesh | SIZE | 25.40 | 0.00 | Karla | SIZE | 5.75 | 0.84 | Tamil Nadu | SIZE | 6.91 | 0.00 |
| | AGE | −0.54 | 0.09 | | AGE | −0.27 | 0.75 | | AGE | −0.32 | 0.00 |
| | C | −81.71 | 0.00 | | C | −6.06 | 0.81 | | C | −13.71 | 0.07 |
| | R-squared | | 0.82 | | R-squared | | 0.78 | | R-squared | | 0.59 |
| | Adjusted R-squared | | 0.78 | | Adjusted R-squared | | 0.71 | | Adjusted R-squared | | 0.51 |
| | Prob(F-statistic) | | 0.00 | | Prob(F-statistic) | | 0.00 | | Prob(F-statistic) | | 0.00 |
| | Durbin-Watson stat | | 2.18 | | Durbin-Watson stat | | 1.87 | | Durbin-Watson stat | | 1.64 |
| | Random Effect | | | | Fixed Effect | | | | Fixed Effect | | |
| | Variable | Coefficient | Prob. | | Variable | Coefficient | Prob. | | Variable | Coefficient | Prob. |
| | FP | −10.76 | 0.07 | | FP | −11.19 | 0.14 | | FP | −21.74 | 0.00 |
| | IP | 1.23 | 0.81 | | IP | 13.51 | 0.01 | | IP | 28.15 | 0.00 |
| | LEVE | −1.46 | 0.00 | | LEVE | −0.14 | 0.18 | | LEVE | 0.01 | 0.85 |
| Dadra & Nagar Haveli | SIZE | 0.15 | 0.89 | Madhya Pradesh | SIZE | 7.07 | 0.12 | Telangana | SIZE | 9.30 | 0.00 |
| | AGE | −0.03 | 0.65 | | AGE | −1.02 | 0.00 | | AGE | −0.63 | 0.00 |
| | C | 9.16 | 0.04 | | C | 6.42 | 0.57 | | C | −15.86 | 0.04 |
| | R-squared | | 0.35 | | R-squared | | 0.41 | | R-squared | | 0.57 |
| | Adjusted R-squared | | 0.30 | | Adjusted R-squared | | 0.29 | | Adjusted R-squared | | 0.49 |
| | Prob(F-statistic) | | 0.00 | | Prob(F-statistic) | | 0.00 | | Prob(F-statistic) | | 0.00 |
| | Durbin-Watson stat | | 1.65 | | Durbin-Watson stat | | 1.85 | | Durbin-Watson stat | | 1.92 |

**Table 7.** *Cont.*

| | | Fixed Effect | | | | Fixed Effect | | | | Random Effect | |
|---|---|---|---|---|---|---|---|---|---|---|---|
| | Variable | Coefficient | Prob. | | Variable | Coefficient | Prob. | | Variable | Coefficient | Prob. |
| | FP | −10.93 | 0.00 | | FP | −0.12 | 0.00 | | FP | −20.34 | 0.00 |
| | IP | 19.00 | 0.00 | | IP | 0.11 | 0.00 | | IP | 13.14 | 0.00 |
| | LEVE | −27.68 | 0.00 | | LEVE | 0.00 | 0.00 | | LEVE | −0.96 | 0.01 |
| Gujarat | D(SIZE) | 11.09 | 0.00 | Maharashtra | SIZE | 0.07 | 0.00 | Uttar Pradesh | SIZE | −0.17 | 0.81 |
| | AGE | 0.09 | 0.30 | | AGE | 0.00 | 0.90 | | AGE | −0.05 | 0.13 |
| | C | 16.85 | 0.00 | | C | 1.61 | 0.02 | | C | 8.34 | 0.02 |
| | R-squared | | 0.73 | | R-squared | | 0.60 | | R-squared | | 0.19 |
| | Adjusted R-squared | | 0.67 | | Adjusted R-squared | | 0.53 | | Adjusted R-squared | | 0.17 |
| | Prob(F-statistic) | | 0 | | Prob(F-statistic) | | 0.00 | | Prob(F-statistic) | | 0.00 |
| | Durbin-Watson stat | | 1.50 | | Durbin-Watson stat | | 1.72 | | Durbin-Watson stat | | 1.70 |
| | | Fixed Effect | | | | Random Effect | | | | Random Effect | |
| | Variable | Coefficient | Prob. | | Variable | Coefficient | Prob. | | Variable | Coefficient | Prob. |
| | FP | −25.03 | 0.00 | | D(FP) | −3.26 | 0.15 | | D(FP) | −13.78 | 0.00 |
| | IP | 24.44 | 0.00 | | IP | 4.86 | 0.06 | | D(IP) | 10.82 | 0.00 |
| | LEVE | −0.11 | 0.07 | | LEVE | −1.00 | 0.00 | | LEVE | −0.06 | 0.27 |
| Haryana | SIZE | 0.28 | 0.94 | NCT of Delhi | SIZE | 0.71 | 0.71 | West Bengal | SIZE | −0.05 | 0.88 |
| | AGE | −0.31 | 0.05 | | AGE | 0.09 | 0.40 | | AGE | 0.01 | 0.21 |
| | C | 11.64 | 0.26 | | C | −3.79 | 0.51 | | C | −0.90 | 0.44 |
| | R-squared | | 0.59 | | R-squared | | 0.61 | | R-squared | | 0.10 |
| | Adjusted R-squared | | 0.51 | | Adjusted R-squared | | 0.52 | | Adjusted R-squared | | 0.09 |
| | Prob(F-statistic) | | 0.00 | | Prob(F-statistic) | | 0.00 | | Prob(F-statistic) | | 0.00 |
| | Durbin-Watson stat | | 1.73 | | Durbin-Watson stat | | 1.68 | | Durbin-Watson stat | | 2.14 |

**Table 7.** *Cont.*

| | Pooled | | | | Fixed Effect | | | | Fixed Effect | | |
|---|---|---|---|---|---|---|---|---|---|---|---|
| | **Variable** | **Coefficient** | **Prob.** | | **Variable** | **Coefficient** | **Prob.** | | **Variable** | **Coefficient** | **Prob.** |
| | | | | | | | | | ROTA(−1) | 0.14 | 0.05 |
| | FP | −0.18 | 0.09 | | FP | −20.26 | 0.00 | | FP | −7.60 | 0.05 |
| | IP | 0.18 | 0.03 | | IP | 13.99 | 0.00 | | IP | −1.39 | 0.76 |
| Himachal Pradesh | LEVE | 0.00 | 0.14 | Odisha | LEVE | −0.30 | 0.29 | Rajasthan | LEVE | −1.16 | 0.07 |
| | SIZE | 0.00 | 0.99 | | SIZE | 3.31 | 0.09 | | SIZE | −0.60 | 0.87 |
| | AGE | 0.00 | 0.88 | | AGE | −0.13 | 0.16 | | AGE | 0.10 | 0.61 |
| | C | 1.56 | 0.00 | | C | −2.78 | 0.64 | | C | 7.42 | 0.39 |
| | R-squared | | 0.24 | | R-squared | | 0.23 | | R-squared | | 0.79 |
| | Adjusted R-squared | | 0.16 | | Adjusted R-squared | | 0.18 | | Adjusted R-squared | | 0.74 |
| | Prob(F-statistic) | | 0.01 | | Prob(F-statistic) | | 0.00 | | Prob(F-statistic) | | 0.00 |
| | Durbin-Watson stat | | 1.64 | | Durbin-Watson stat | | 1.91 | | Durbin-Watson stat | | 1.69 |
| | Random Effect | | | | Fixed Effect | | | | Fixed Effect | | |
| | Variable | Coefficient | Prob. | | Variable | Coefficient | Prob. | | Variable | Coefficient | Prob. |
| | FP | −26.34 | 0.00 | | D(FP) | −25.44 | 0.00 | | D(FP) | −11.52 | 0.00 |
| | IP | 19.31 | 0.00 | | IP | 9.91 | 0.21 | | IP | 14.65 | 0.00 |
| | LEVE | −0.02 | 0.87 | | LEVE | −0.02 | 0.50 | | D(LEV) | −0.09 | 0.00 |
| Karnataka | SIZE | −0.28 | 0.81 | Punjab Punjab | SIZE | 5.25 | 0.32 | All states | D(SIZE) | 17.82 | 0.00 |
| | AGE | −0.11 | 0.09 | | AGE | 0.30 | 0.18 | | AGE | −0.04 | 0.39 |
| | C | 8.13 | 0.10 | | C | −32.31 | 0.07 | | C | -3.43 | 0.08 |
| | R-squared | | 0.21 | | R-squared | | 0.63 | | R-squared | | 0.6832 |
| | Adjusted R-squared | | 0.19 | | Adjusted R-squared | | 0.54 | | Adjusted R-squared | | 0.6191 |
| | Prob(F-statistic) | | 0.00 | | Prob(F-statistic) | | 0.00 | | Prob(F-statistic) | | 0 |
| | Durbin-Watson stat | | 1.49 | | Durbin-Watson stat | | 2.32 | | Durbin-Watson stat | | 1.6646 |

The regression results in Table 7 demonstrate the impact of the working capital policies on the performance of manufacturing firms operating across Indian states. Findings reveal that the working capital financing policy negatively and significantly impacts the profitability of manufacturing firms located in Andhra Pradesh, Gujarat, Haryana, Karnataka, Maharashtra, Odisha, Punjab, Tamil Nadu, Telangana, Uttar Pradesh, and West Bengal (P.V < 0.05). This implies that when the ratio of current liabilities to total assets increases, firms' profitability measured by return on assets decreases. Furthermore, it is shown in Table 7 that working capital investing policy has a positive and significant impact on the performance of firms doing business across all Indian states, except companies that are doing business in Rajasthan. This result indicates that when current assets to total assets ratio increase, profitability of Indian manufacturing firms (measured by return on assets) increases. This can be applied when firms invest more in current assets, in line with the conservative investment policy. Moreover, companies that are working in Dadra and Nagar Haveli, Kerala, NCT of Delhi, and Punjab in which investing policy insignificantly affect firms' profitability.

Leverage negatively and significantly impacts return on assets of manufacturing companies that are located in Dadra and Nagar Haveli, Gujarat, Maharashtra, NCT of Delhi, and Uttar Pradesh. Furthermore, the results of the whole sample show that Indian manufacturing companies get positively and significantly impacted by leverage. This study also found that firms' size positively affects firms working in all states except for companies which are operating in Karnataka and Rajasthan. Results in Table 7 show that age has an insignificant impact on firms' performance working in all states covered in this study except those companies which are working in Madhya Pradesh, Tamil Nadu, and Telangana.

## 5. Results Discussion

Results in Table 3 revealed that the mean values of financing policy ratio for all manufacturing companies located at different states range between 0.34 and 0.44. Furthermore, the mean value of the same ratio for the whole sample is 0.38. These results indicate that approximately one third of total assets in Indian firms are financed by short-term debt. Adam et al. [44] and Weinraub and Visscher [31] argue that a low ratio (current liabilities/total assets), i.e., less than 50%, implies that firms are following relatively conservative financing policy. This means that Indian manufacturing firms are adopting conservative financing policy, which is expected to have a positive effect on the profitability of the firms and their financial sustainability. Pestonji and Wichitsathian [19] in their study conducted in Iran found out that conservative financing policy has a positive relationship with firms' profitability. When a firm finances its current assets and part of its fixed assets by short-term debt, it increases the profitability. Results also found that some manufacturing companies working in Maharashtra, Telangana, NCT of Delhi, West Bengal, Karnataka, and Madhya Pradesh states are following extreme financing policy as the ratio (current liabilities/total assets) of these companies is below 0.02. This result could be explained by the fact that these companies might have faced difficulties in getting short-term funding from outsource, and these companies would have enough internal sources for financing their investments and had financed their investment by issuing shares or getting long-term dept.

Regarding the investment policy, results in Table 3 showed that the mean values of the investment policy ratio for all manufacturing companies located at different states range between 0.43 and 0.61. Furthermore, the mean value of the investment policy ratio for the whole sample is 0.51. These results imply that Indian manufacturing companies invest more in current assets and less in fixed assets. Adam et al. [44] and Weinraub and Visscher [31] believe that a high ratio (current assets/total assets), i.e., more than 50%, means that firms are following relatively conservative policy. Thus, it can be said that Indian manufacturing companies are following relatively conservative policy. The study explains these results by the fact that Indian manufacturing firms are not risk takers; thus, they invest in current assets more than in fixed assets.

Results in Table 7 revealed that the conservative financing policy followed by Indian manufacturing firms negatively and significantly affected their profitability. This result implies that when current-liabilities/total-assets ratio increases, the profitability of Indian manufacturing firms decreases. Therefore, using a low level of debt in financing assets leads to higher profitability and helps firms to attain financial sustainability. These results imply that $H_{01}$ is accepted. This study suggests that adopting more conservative working capital financing policy is appropriate for Indian manufacturing firms in order to have better financial sustainability. This result is consistent with Afza and Nazir [28] in Pakistan, Weinraub and Visscher [31], and Vahid et al. [18] in Iran who advocated that financing policy has a negative impact on firms' performance. These findings are in contrast with the theory that states that aggressive financing policy increases firms' profitability as long as the cost of short-term debt is lower than the long-term debt [24,28].

On the contrary, the study found that the conservative investing policy followed by Indian manufacturing companies positively and significantly affected their profitability and therefore their financial sustainability. This result implies that more investment in current assets increases profitability, which indicates that financial managers can create value for shareholders and attain more financial sustainability if they follow a conservative investment policy. However, it is theoretically believed that aggressive working capital investment policy increases profitability, in practice, and conservative investment policy results in higher profit. The reason behind the high level of investment in current assets is that manufacturing companies need to expand both inventories and trade credit to their customers, while service companies do not need that number of inventories and accounts receivables. This result is in line with that of Vahid et al. [18] in Iran. This finding disagrees with the results that were found by Tauringana and Afrifa [50] and Al-Shubiri [17] who argued that the working capital investing policy has a negative impact on firms' profitability in the UK.

Concerning controlling variables, it was found that leverage has a negative and significant effect on the return on assets and hence on performance sustainability of Indian manufacturing companies that are located in Dadra and Nagar Haveli, Gujarat, Maharashtra, NCT of Delhi, and Uttar Pradesh. This result indicates that when leverage ratio increases, the profitability of Indian manufacturing firms decreases. This result is consistent with Vahid et al. [18] in Iran, [50] in UK, Abuzayed [23] in Jordan, Teruel and Solan [24] in Spain, Sharma and Kumar [51] in India, and Raheman and Nasr [52], Khan et al. [53] in Pakistan. However, these results contradict with Lyngstadaas and Berg [25] in Norway and Afrifa [26] in the UK. Furthermore, this study also found that firms' size positively affects the profitability of Indian manufacturing firms and thus the financial sustainability. This result means that when size of a firm increases, the return on assets of Indian manufacturing firms increases. This result is consistent with Vahid et al. [18] and Deloof [4] in Belgian, Khan et al. [50] in Pakistan, Abuzayed [23] in Jordan, Abuzayed Teruel and Solan [24] in Spain, Lyngstadaas and Berg [25] in Norway, and Afrifa [26] in UK, and it contradicts Tauringana and Afrifa [50], Sharma and Kumar [51] in India, and Raheman and Nasr [52] in Pakistan. Finally, results in Table 7 show that age does not have any significant effect on the profitability of Indian manufacturing firms' profitability. This result is in contrast with Afrifa [26] in the UK and Khan et al. [53] in Pakistan. Table 8 compares the results of this study with the results that were found by previous studies.

Overall, this study argues regulators, policymakers, investors, and financial managers in Indian manufacturing companies to favor conservative working capital financing and investment policies for to be more profitable and attain financial sustainability.

**Table 8.** Summary results obtained by some authors.

| Authors | | Firms' Profitability | | | | |
|---|---|---|---|---|---|---|
| | | Positive | Negative | Study's Result | Consistent | Contradicted |
| Afza and Nazir [28] in Pakistan | Financing policy | | ✓ | Negative | ✓ | |
| Weinraub and Visscher [31] in Iran | | | ✓ | | ✓ | |
| Vahid et al. [18] in Iran | | | ✓ | | ✓ | |
| Khan et al. [53] in Pakistan | | | ✓ | | ✓ | |
| Vahid et al. [18] in Iran | Investing policy | ✓ | | Positive | | ✓ |
| Khan et al. [53] in Pakistan | | ✓ | | | | ✓ |
| Tauringana and Afrifa [50] | | | ✓ | | ✓ | |
| ALShubiri [17] | | | ✓ | | ✓ | |

## 6. Conclusions

In order to answer the two posed questions, the study attempted to find out the type of working capital policies adopted by Indian firms across all states and evaluate the impact of working capital management policies on firms' profitability across all Indian states. India has 36 states and union territories, and out of them, 17 states and union territories were included in this study due to the methodology process followed in the study. For achieving the objectives of this research, the study extracted data from Prowess IQ database for the period from 2011 to 2017 for 2181 manufacturing companies. After screening the data, some companies were excluded from the sample, and therefore, the final sample consists of 829 companies. The dependent variable, firms' profitability, is measured by ROA, and the independent variable, working capital policy, is measured by financing and investing policies. The model was controlled by leverage, firm's size, and age. The study predicted that working capital financing and investing policies negatively and significantly impact firms' profitability. A panel fixed- and random-effect model approach was adopted to estimate the results.

Regarding the first research question about the type of followed working capital policy, results revealed that manufacturing companies across Indian states adopt conservative financing and investing policies. Concerning the second research question about the impact of followed working capital policies on the profitability, results showed that the conservative investment policy followed by Indian manufacturing firms positively affected the profitability of Indian manufacturing firms and thus on their financial sustainability; this result went against the predicted result of the study. On the other hand, the conservative financing policy negatively affected firms' profitability measured by return on assets, and therefore on the ability in attaining financial sustainability, this result went in line with the predicted result of the study.

The present study provides potentially useful implications for regulators, policymakers, investors, and financial managers in Indian manufacturing companies which highlight the importance of working capital management policies to the profitability and financial sustainability of firms. Thus, regulators, policymakers, investors, and financial managers should focus on working capital management policies, because when they are carefully formulated, firms are unlikely to be prone to liquidity risk and financial sustainability failure. Furthermore, financial managers are advised to follow a conservative investment and financing policy which are effective and efficient in boosting firms' profitability. Therefore, firms should invest more in current assets, in line with the conservative investment policy because manufacturing companies need to expand both inventories and trade credit to their customers. This strategy eliminates adverse risks and its associated costs, to a larger extent, and ultimately leads to an improvement in profits. Moreover, financial managers are advised to favor a low level of debt in financing assets which leads to higher profitability. Finally, researchers and academicians have to shed more insight on the importance of

working capital management policies to firms' profitability. Before taking a decision based on the results of the study, some limitations have to be taken into consideration. Firstly, the study is limited only to manufacturing companies, making it difficult to generalize the findings on the other sectors. Therefore, studies with diversified samples are recommended. Secondly, this study is only based on secondary data in comparison to other research areas in finance field. Thus, further studies on the same could be conducted using a questionnaire survey. Another limitation is the lack of access to updated data beyond 2017. However, it is believed that data from 2008 to 2017 are enough for getting robust results. Further, the data of 2018 and 2019 would not affect the results.

Notes:

1. The study targeted all Indian manufacturing companies across 36 states and union territories which are listed on Bombay Stock exchange.
2. To overcome the problem of heteroscedasticity ROA and FP were transformed by logarithm in the case of Himachal Pradesh, Madhya Pradesh, and Maharashtra data.
3. Differencing the variable by using (d) was used for the following variables: FP in the case of NCT of Delhi and Punjab, firms' size in the case of Gujarat and FP, and LEVE and firm size in the case of the whole sample.
4. To tackle the presence of autocorrelation ROA was lagged in the case of Rajasthan and FP, IP, and ROA in the case of West Bengal.
5. Breusch–Pagan–Godfreytest was applied to check the presence of hetroscedacity as shown in Table 9.

**Table 9.** Breusch–Pagan–Godfreytest.

| State | Prob. Chi-Square | State | Prob. Chi-Square | State | Prob. Chi-Square |
|---|---|---|---|---|---|
| Andhra Pradesh | 0.7135 | Kerla | 0.0522 | Rajasthan | 0.2325 |
| Dadra and Nagar Haveli | 0.4019 | Madhya Pradesh | 0.3975 | Tamil Nadu | 0.0596 |
| Gujarat | 0.0853 | Maharashtra | 0.7254 | Telangana | 0.4509 |
| Haryana | 0.864 | Nct of Delhi | 0.1402 | Uttar Pradesh | 0.0644 |
| Himachal Pradesh | 0.1152 | Odisha | 0.0648 | West Bengal | 0.2476 |
| Karnataka | 0.5357 | Punjab | 0.4278 | all states | 0.2845 |

**Author Contributions:** Conceptualization, N.H.SF. and E.M.A.-M.; methodology, F.A.A.; software, F.A.A. and W.M.A.; validation, N.H.SF., N.A.M.S. and S.A.H.; formal analysis N.H.SF. and F.A.A.; investigation, E.M.A.-M. and W.M.A.; resources, N.A.M.S. and S.A.H.; data curation, F.A.A. and E.M.A.-M.; writing—original draft preparation, N.H.SF.; writing—review and editing, N.H.SF., E.M.A.-M., N.A.M.S., W.M.A., F.A.A., S.A.H.; visualization, N.H.SF.; supervision, F.A.A.; project administration, N.H.SF. and W.M.A.; funding acquisition, E.M.A.-M. and N.A.M.S. All authors have read and agreed to the published version of the manuscript.

**Funding:** This research received no external funding.

**Institutional Review Board Statement:** Not applicable.

**Informed Consent Statement:** Not applicable.

**Data Availability Statement:** The data of this study was obtained from ProwessIQ. ProwessIQ is a database of about 38,000 Indian companies. https://prowessiq.cmie.com (accessed on 9 April 2021).

**Conflicts of Interest:** The authors declare no conflict of interest.

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
