# Peer review of "Working Capital Management Policies in Indian Listed Firms: A State-Wise Analysis"

_sustainability, doi:10.3390/su13084516_

Round 1

Reviewer 1 Report

Dear authors,

Your article is very interesting, and I am grateful for the opportunity to read it.

Reading the text, I found some elements you should change and I think that would improve your article.

The abstract clearly defines the subject and method of the research, as well as the results.

Introduction - The introduction should contain general information and research background. In my opinion, the analysis of the subject should be expanded - a high level of generality does not allow to introduce readers to the topic well. I would suggest that there you propose your vision of the text, its goals and ways of working with the topic. So, my proposition is to add explicit the research goals, research questions (if you have), describe the outline and framework of your research and define the directions of your research.

The literature review contains important information regarding the current state of knowledge. The content should be the starting point for further analysis.

The 3rd chapter – I don’t understand why you put it in this place. I think this is the part of the results of your research, so please try to unify it in one chapter called „Results”.

The methodology is well prepared and also well presented.

The 5th chapter. I think it is small misunderstood. This place is reserved for a chapter „Results”. You should remodel it and use this chapter to show your work and its output. So I think it won’t be tough to do it but the effect you will achieve allow you to clarify your article and well present the results.

Discussion - To increase the relevance of the results, the discussion section should cover the differences and similarities between your findings and that of other scholars. You Should create a chapter „Discussion” and extend information about the results of others research. For me, this part (it means the last 2 paragraphs in this chapter) is too limited,

Conclusions: The concluding part should be a short summary of the goals, methods and findings of the article. There is too modest reference to your assumptions, your thesis/hypotheses or research questions. This is where you should show references to your research and all formal aspects of your article. At the beginning and at the end, a description of research questions and research hypotheses should be included. In my opinion, the goals should be presented at the beginning and explained at the end. Conclusions should relate to each of your goals, not just the results of your research.

I don’t understand the way of presentation the sources at the beginning of the phrases:

[13], [4] and [15] argued that conservative working capital investment policy results in a huge amount of investment in current assets.

[21] argued that there is a negative relationship between aggressive investment policy and Tobin’s Q, while

I think you should use names of authors, because this form is not good.

Summarizing. I think you have to remodel the structure of your article because the current structure is not clear and that way you lose the value of your work. You should to extend the discussion part and better describe formal aspects of your article (thesis/questions, goals etc.). 

Despite my comments, I found your article very interesting and I appreciate your work.

Good luck!

Author Response

I am addressing herewith all Frist reviewer's comments of the manuscript entitled “Working Capital Management Policies in Indian Listed Firms: A State Wise Analysis” manuscript ID 1119088.

Reviewer 2 Report

The main aim of this paper is to evaluate the impact of financing and investment working capital policies on firms’ profitability.

The paper has to be checked for English grammar and typos. Example, in Abstract Section: “gab”, “pervious”.

Regarding the reference to Quansah in the Introduction Section, the argumentation is not well understandable. Explain in more detail as these lines are confusing: “[19] argued that if firms adopt an aggressive working capital policy, they should balance it with a conservative working capital policy”.

The division of the Introduction section and the Literature review section is not clear. I recommend keeping Introduction section to basically justify your work and comment on how you are going to implement it. Several arguments are repeated in both sections.

There are sentences literally repeated throughout the manuscript, for example on page 2 and page 5. “For instance, the impact of a change in food prices…”

The authors compare and sufficiently reference the results obtained by other authors in other countries. Nevertheless, a summary table of the results obtained by the different authors would be appreciated.

The authors do not comment on the results of Table 4 with respect to the VIF values and the limit to consider whether multicollinearity may exist. Is it 10, 5 or for example 2.5?.

The authors should further explain and clarify the first paragraph on page 9 and how these results lead to the conclusion that “Results reveled that companies working in different states adopt a conservative financing and investing policies”.

The authors should make a greater effort to relate the subject of their study to sustainability.

Hope it helps. Best wishes

Author Response

I am addressing herewith all second reviewer's comments of the manuscript entitled “Working Capital Management Policies in Indian Listed Firms: A State Wise Analysis” manuscript ID 1119088 + Change of Authorship Form

Round 2

Reviewer 2 Report

Suggested changes have been modified.

A comma should be removed after citation 45 in the variable control section:

The study control variables are drawn from prior research on working capital management [e.g., 26, 28, 4, 45, , 8, 42, 43, 44]
